Review

 

Subject Area:
biotechnology

Keywords:
metabolic engineering, biofuels, cell factory, *Saccharomyces cerevisiae*

Author for correspondence:
Jens Nielsen
e-mail: nielsenj@chalmers.se

# Engineering *Saccharomyces cerevisiae* cells for production of fatty acid-derived biofuels and chemicals

Yating Hu[1,2], Zhiwei Zhu[1,2], Jens Nielsen[1,2,3,4] and Verena Siewers[1,2]

[1]Department of Biology and Biological Engineering, and [2]Novo Nordisk Foundation Center for Biosustainability, Chalmers University of Technology, 41296 Gothenburg, Sweden
[3]Novo Nordisk Foundation Center for Biosustainability, Technical University of Denmark, 2800 Kgs Lyngby, Denmark
[4]BioInnovation Institute, Ole Måløes Vej, 2200 Copenhagen N, Denmark

JN, 0000-0002-9955-6003

The yeast *Saccharomyces cerevisiae* is a widely used cell factory for the production of fuels and chemicals, in particular ethanol, a biofuel produced in large quantities. With a need for high-energy-density fuels for jets and heavy trucks, there is, however, much interest in the biobased production of hydrocarbons that can be derived from fatty acids. Fatty acids also serve as precursors to a number of oleochemicals and hence provide interesting platform chemicals. Here, we review the recent strategies applied to metabolic engineering of *S. cerevisiae* for the production of fatty acid-derived biofuels and for improvement of the titre, rate and yield (TRY). This includes, for instance, redirection of the flux towards fatty acids through engineering of the central carbon metabolism, balancing the redox power and varying the chain length of fatty acids by enzyme engineering. We also discuss the challenges that currently hinder further TRY improvements and the potential solutions in order to meet the requirements for commercial application.

## 1. Introduction

The growing demand for liquid transport fuels alongside concerns about climate change caused by greenhouse gas emissions from the use of fossil fuels has become one of the greatest challenges for modern society. The generation of biofuels from biomass is a sustainable solution that could substantially decrease the usage of fossil fuels. In past decades, various policies have been established to stimulate global biofuel production [1]. Ethanol is the predominant biofuel and has been used in Europe and the USA since the 1900s [2], and global ethanol production is expected to expand from 120 billion litres in 2017 to 131 billion litres by 2027 [3]. Even though the use of ethanol allows a reduction in $CO_2$ emissions of up to 80% compared with using petrol, its low energy density and hygroscopicity have become an obstacle for its wider application [4,5]. Fatty acid-derived biofuels, such as fatty alcohols and hydrocarbons, have been proposed as an option for use within transport sectors where there is a need for high-density fuels, e.g. aviation and heavy trucks [6]. Fatty acids are naturally produced by cells for both chemical and energy storage functions. Therefore, producing fatty acid-derived biofuels by microorganisms as an alternative biofuel production method has drawn more and more attention, and significant progress has been achieved owing to the dramatic advances in biotechnology.

The diversity of microorganisms, such as fungi, bacteria and algae, allows for the usage of a wider range of substrates, which enables expansion from the use of solely starch-based agricultural products to lignocellulosic biomass waste, $CO_2$, methane, etc. [7]. Bioethanol is currently mainly derived from corn starch or cane sugar, which are fermented by microorganisms, such as

yeast [7–9]. *Yarrowia lipolytica*, a model oleaginous yeast, and other oleaginous yeasts are being established as promising platforms for biofuel generation [10]. However, owing to its essential role in bioethanol production, the yeast *Saccharomyces cerevisiae* has become one of the most intensively industrially applied cell factories, offering the possibility of alternative advanced biofuel production based on existing infrastructures and assets without any extra facility costs. Furthermore, the ease of genetic manipulation and its robustness and tolerance towards harsh conditions in industrial production also contribute to the popularity of yeast as a platform to generate various chemicals [11,12]. We will therefore focus on *S. cerevisiae* in this review.

As the major component of cell membranes, fatty acids and their metabolism have been comprehensively studied in yeast [13]. Metabolic engineering strategies, such as blocking competing pathways, increasing precursor supply and balancing cofactor regeneration in the cell, have been applied to establish optimal native and heterologous pathways for sustainable production of fatty acids [14,15]. In addition, several advanced biofuels including alkanes, fatty alcohols and fatty acid ethyl esters (FAEEs), for which fatty acids serve as precursors, were successfully generated in yeast [16–19]. Progress in synthetic and systems biology has also enabled the construction of yeast strains that produce fatty acids and fatty alcohols with a chain length that cells lack the capacity to generate naturally. Here, we will highlight the major contributions to the production of fatty acid-derived biofuels and chemicals through different metabolic engineering strategies in yeast, and point to the major challenges and directions for future laboratory-scale studies and industrial applications.

## 2. Engineering of central carbon metabolism

### 2.1. Enhancing the precursor supply for fatty acid synthesis

Fatty acids with a long aliphatic chain are naturally produced by yeast in either their saturated or monounsaturated form. Acetyl-coenzyme A (acetyl-CoA) as the main C2 metabolite is the essential building block for fatty acid synthesis (FAS). Although acetyl-CoA is involved in the metabolic network of *S. cerevisiae* in the cytosol, nucleus, peroxisome and mitochondrion, it is not transported freely across membranes in the absence of the carrier carnitine. However, different shuttle mechanisms exist. The major substrate for de novo FAS in yeast is cytosolic acetyl-CoA, which is generated from pyruvate via three reactions catalysed by pyruvate decarboxylase (Pdc), acetaldehyde dehydrogenase (ALD) and acetyl-CoA synthetase (ACS) (figure 1). Acetaldehyde, as the intermediate generated from pyruvate, is also the precursor for ethanol production, which is an undesired by-product when aiming for high-yield fatty acid production. Thus, many efforts have been directed towards improving the cytosolic acetyl-CoA pool. Alcohol dehydrogenase (ADH) genes were deleted to prevent the conversion from acetaldehyde to ethanol and/or ALD and endogenous or heterologous ACS were overexpressed to enhance the carbon flux to

acetyl-CoA [20]. However, ethanol production is hard to eliminate in yeast by simply deleting ADH genes, as there is a large number of promiscuous ADHs that could catalyse the reaction to generate ethanol and many of these are also involved in other important reactions within the cell [21,22]. In order to overcome this problem, ethanol formation was inhibited by the elimination of all three Pdc enzymes (Pdc1, Pdc5 and Pdc6) [23]. Nevertheless, such a Pdc-negative strain is unable to grow in excess glucose possibly because of repression of the respiratory metabolism and a deficiency in cytosolic C2 supply. Pronk and co-workers [23] solved this problem by evolving the Pdc-deficient strain and succeeded in obtaining a C2-independent Pdc-negative strain which could also grow in excess glucose. Later studies revealed that the adaptation mechanism was associated with an internal deletion in a transcriptional regulator, Mth1, which is involved in glucose sensing in yeast. Such a mutated version of Mth1 reduced the glucose influx and thus resulted in decreased repression of respiration in the evolved strain [24,25].

Although ethanol synthesis is blocked in a Pdc-negative strain, this also decreases the amount of cytosolic acetyl units that serve as precursors for various downstream products including fatty acids. A route relying on mitochondrial Ach1, the CoA transferase hydrolysing mitochondrial acetyl-CoA to acetate that enters the cytosol to provide the C2 unit for cytoplasmic acetyl-CoA synthesis, was shown to compensate for the lack of cytosolic acetyl-CoA synthesis [26]. However, this strategy is restricted by the limited mitochondrial acetyl-CoA supply owing to the stringent regulation of the pyruvate dehydrogenase (PDH) complex and this route cannot function under glucose-repressed conditions (figure 1). Alternatively, a heterologous PDH complex from *Enterococcus faecalis* was expressed in an ACS-deficient yeast strain and shown to fully complement the cytosolic acetyl-CoA supply [27]. Other attempts implemented in yeast to benefit cytosolic acetyl-CoA supply as well include introducing heterologous acetylating ALD (A-Ald), pyruvate-formate lyase (PFL), a phosphoketolase (PHK) pathway or pyruvate oxidase (figure 1) [27–30]. Recently, our group reprogrammed the yeast central metabolism to demonstrate a feasible strategy for industrial production of fatty acids with high titre and yield [31]. A heterologous ATP citrate lyase (ACL) was overexpressed to provide cytosolic acetyl-CoA in an engineered fatty acid overproducing strain (figure 1). Based on that, the three Pdc genes were deleted to abolish ethanol production. After adaptive laboratory evolution (ALE) in glucose, the evolved strain exhibited a pure lipogenesis metabolism, resulting in a great improvement in fatty acid production. Using a similar strategy, expression of ACL from *Y. lipolytica*, downregulation of malate synthase (Mls1) and deletion of glycerol-3-phosphate dehydrogenase (Gpd1) were carried out in *S. cerevisiae*—the latter two being key enzymes involved in competing with FAS for carbon flux—leading to a 70% improvement in free fatty acid production [32].

The conversion of acetyl-CoA to malonyl-CoA via acetyl-coenzyme A carboxylase (ACCase) encoded by *ACC1* is the first committed and rate-limiting step in de novo FAS in yeast. Increasing malonyl-CoA supply is a promising strategy that benefits fatty acid production. In order to break through the limitation of low efficiency of this reaction, the Snf1-dependent phosphorylation of ACCase was—at least

royalsocietypublishing.org/journal/rsob    Open Biol. 9: 190049

royalsocietypublishing.org/journal/rsob    Open Biol. 9: 190049

**Figure 1.** Fatty acid synthesis (FAS) pathway in *S. cerevisiae* and engineering strategies of the central carbon metabolism for increasing fatty acid production. The solid arrows and dashed arrows represent single catalytic steps and multiple catalytic steps, respectively. The arrows in blue represent the heterologous pathway that was introduced into yeast. G6P, glucose-6-phosphate; F6P, fructose-6-phosphate; DHAP, dihydroxyacetone phosphate; G3P, glyceraldehyde-3-phosphate; VLCFA, very-long-chain fatty acid; Pgi, phosphoglucose isomerase; Zwf, D-glucose-6-phosphate dehydrogenase; Gnd, phosphogluconate dehydrogenase; Gpd, glycerol-3-phosphate dehydrogenase; GAPN, glyceraldehyde-3-phosphate dehydrogenase; Pdc, pyruvate decarboxylase; Pdh, pyruvate dehydrogenase; A-Ald, acetylating acetaldehyde dehydrogenase; Pfl, pyruvate-formate lyase; Adh, alcohol dehydrogenase; Ald, acetaldehyde dehydrogenase; Acs, acetyl-CoA synthetase; ACL, ATP citrate lyase; Acc, acetyl-coenzyme A carboxylase; TE, thioesterase; Elo1/2/3, fatty acid elongase; Faa1/4, fatty acyl-CoA synthetases; AT, acetyl transferase; MPT, malonyl/palmitoyl transferase; KS, ketoacyl synthase; KR, ketoacyl reductase; CH, dehydratase; ER, enoyl reductase.

partially—abolished by introducing two mutations corresponding to Ser1157 and Ser659 of *ACC1* ($ACC1^{S1157A,S659A}$) [33]. An even more efficient catalytic activity was observed when S686A was introduced into the double mutant *ACC1* ($ACC1^{S1157A,S659A,S686A}$) [34]. The resulting higher ratio of malonyl-CoA/acetyl-CoA shifts production towards C18 fatty acids, and the overexpression of either wild-type *ACC1* or mutant *ACC1* can lead to an improvement in fatty acid production [16,35].

## 2.2. Balancing cofactor supply and reducing power

Reducing power is an essential element involved in many metabolite conversions. NADH and NADPH, the two major reducing equivalents in yeast, play distinct functions in the cell, i.e. NADH predominantly participates in catabolic reactions and NADPH is mainly required for anabolic reactions [36]. The ratios of the two pyridine nucleotide cofactor systems NADH/NAD$^+$ and NADPH/NADP$^+$ are vital for the determination of the cellular redox status and the formation of various metabolites.

Cellular NADPH, the essential reducing equivalent for fatty acid formation and other metabolic conversions in yeast, is predominantly generated from the pentose phosphate (PP) pathway (figure 1). D-Glucose-6-phosphate (G6P) is oxidized through D-glucose-6-phosphate dehydrogenase (G6PDH) encoded by *ZWF1*, which is the first rate-limiting step in the PP pathway. Then, ribulose-5-phosphate is generated through 6-phosphogluconate dehydrogenase

(6PGDH), encoded by *GND1* and *GND2*, thereby yielding two molecules of NADPH. For de novo FAS in yeast, two molecules of NADPH are required as cofactors for each cycle of elongation. Since yeast cells naturally produce excess NADH as the electron carrier, the NADPH supply is often limiting for anabolic reactions. Thus, attempts at metabolic engineering have been made to increase the NADPH supply in the cytosol. To facilitate NADPH regeneration and reduce loss of the carbon source, a non-phosphorylating NADP$^+$-dependent glyceraldehyde-3-phosphate dehydrogenase (GAPN) from *Bacillus cereus* was expressed in a yeast strain carrying a deletion in the *GPD1* gene encoding NAD$^+$-dependent glycerol-3-phosphate dehydrogenase [37] (figure 1). Extra NADPH can also be produced by overexpression of the otherwise mitochondrial malic enzyme (ME), which converts malate into pyruvate in the cytosol [19,38]. The PHK pathway, which uses xylulose-5-phosphate as a precursor, was introduced into *S. cerevisiae* to increase the NADPH supply in the cytosol. The combination of the PP pathway and PHK pathway as an interesting alternative for fatty acid derivative production resulted in improved production of FAEEs [39]. Fine tuning the flux distribution between the PP pathway and glycolysis by overexpression of phosphogluconate dehydrogenase (encoded by *GND1*), transketolase (encoded by *TKL1*) and transaldolase (encoded by *TAL1*) together with downregulation of phosphoglucose isomerase (encoded by *PGI1*) in yeast to provide additional NADPH led to a 28% increase in free fatty acid production [29].

## 3. De novo fatty acid synthesis in yeast

The biosynthesis of fatty acids in yeast can take place in the cytosol and the mitochondria, where it is carried out by a type I FAS and a type II FAS, respectively. Experimental results suggested that the mitochondrial FAS II pathway is the sole source of the octanoic acid required for lipoic acid production. Lipoic acid serves as an essential cofactor for PDH, α-ketoglutarate dehydrogenase and the glycine cleavage system [40,41]. However, the range of fatty acids produced by the mitochondrial FAS II pathway and other potential roles in cellular metabolism are still uncertain [41].

As a type I FAS is responsible for the cytosolic de novo FAS, we will mainly focus on it in this review. The type I FAS in yeast comprises two subunits, α-subunit Fas2 and β-subunit Fas1. Six copies of eight independent functional domains are assembled into an $\alpha_6\beta_6$ molecular complex of 2.6 MDa [42] (figure 1). The yeast FAS is activated by its phosphopantetheinyl transferase domain located at the C-terminus of the α-subunit, and all the reactions occur in the limited space of the $\alpha_6\beta_6$ complex [42]. The yeast FAS initiates the reaction by transferring acetyl primer and malonyl elongation substrate from acetyl-CoA/malonyl-CoA to the acyl carrier protein (ACP) pantetheine arm by the acetyl transferase (AT) and malonyl/palmitoyl transferase (MPT), respectively. The ketoacyl synthase (KS) condenses them to acetoacetyl-ACP in a malonyl decarboxylation reaction, which is considered the first step of the elongation cycle. Subsequently, the β-ketoacyl-ACP is reduced by the ketoacyl reductase (KR) in the α-subunit, followed by a dehydration reaction catalysed by the dehydratase (DH) and the second reduction reaction catalysed by the enoyl reductase (ER) in the β-subunit yielding acyl-ACP [43] (figure 1). The ACP domain, which plays the central role in shuttling intermediates between the active sites in the complex, brings the processed acyl chain back to the KS domain for the next elongation cycle [44]. This repetitive process occurs using malonyl-CoA as the provider of 2C units until the carbon chain length of the fatty acid reaches 16 or 18. The end product will be shuttled by ACP from ER to MPT, where it is transferred to CoA and then released. Previous results showed that the overexpression of native FAS1 and FAS2 in S. cerevisiae could contribute to the fatty acid production as well as introducing heterologous type I or type II FASs [16,19,45,46]. Additionally, in order to overproduce fatty acids in yeast, a common strategy is to overexpress heterologous acyl-ACP or acyl-CoA thioesterases, which can relieve feedback inhibition and increase the fatty acid release [16]. Moreover, the engineering of type I FAS towards short/medium-chain fatty acid (S/MCFA) production has recently attracted attention. A thioesterase from Acinetobacter baylyi ('AcTesA) that has a substrate preference for short/medium-chain acyl-ACP/CoA was embedded into the type I FAS, which benefitted S/MCFA production significantly and led to a 5- to 13-fold increase in S/MCFA production compared with wild-type FAS [47]. Rational modification of (i) the KS domain to restrict chain length elongation, (ii) the MPT domain to reduce the affinity to its substrate malonyl-CoA, and (iii) the AT domain to increase its affinity to acetyl-CoA has succeeded in altering the chain length of fatty acid products resulting in production of extracellular S/MCFAs, mainly hexanoic acid and octanoic acid, of $464 \, \text{mg} \, \text{l}^{-1}$ in total [48]. Very-long-chain fatty acids (VLCFAs) are the precursors for various valuable chemicals and the essential components for yeast cell membrane structures. VLCFA synthesis occurs at the endoplasmic reticulum membrane with distinct enzymes similar to the domains in the FAS system [49]. Of the three fatty acid elongases (equivalent to the KS domain in FAS), Elo1 is responsible for elongation of C12–16 fatty acids to C16–18 fatty acids, while Elo2 and Elo3 are more specifically responsible for the synthesis of up to C22 fatty acid and C26 fatty acid, respectively [50,51] (figure 1). Yeast was successfully engineered for the production of VLCFAs and derived products by the selective modification of the endogenous yeast fatty acid elongation system together with the expression of a heterologous FAS I system from Mycobacterium vaccae [52].

## 4. Fatty acid-derived biofuels and chemicals

### 4.1. Fatty alcohols

Fatty alcohols are important oleochemicals with wide industrial applications ranging from cosmetics to substitutes for petroleum-derived compounds such as biofuels [53]. Fatty alcohols can be generated from fatty acyl-CoAs, fatty acyl-ACPs and fatty acids with fatty aldehydes as the intermediates via the corresponding enzymes fatty acyl-CoA reductase, fatty acyl-ACP reductase and carboxylic acid reductase, respectively [54–56]. These enzymes usually catalyse the first step of the two consecutive reduction steps, i.e. fatty aldehyde formation, followed by the second reaction towards fatty alcohol production via aldehyde reductases/alcohol dehydrogenases (ALRs/ADHs) (figure 2). However, some fatty acyl-CoA/ACP reductases can catalyse the entire four-electron reduction step to generate fatty alcohols directly, for example the well-known FACoAR enzymes from jojoba plant and Arabidopsis thaliana [57]. The heterologous pathways including these enzymes have been successfully introduced into yeast. The production of fatty alcohols in S. cerevisiae was drastically improved by rewiring central metabolic pathways. The production of free fatty acids which served as the precursors for fatty alcohol formation in this strain was improved significantly by deleting fatty acyl-CoA oxidase (encoded by POX1) and fatty acyl-CoA synthetases (encoded by FAA1 and FAA4) (figure 2). Reversal of aldehyde formation was abolished by deleting aldehyde dehydrogenase (encoded by HFD1) [19] (figure 2). Combined with (over)expression of the genes involved in fatty alcohol synthesis, which were in this case fatty acyl-CoA reductase FaCoAR from Marinobacter aquaeolei VT8, carboxylic acid reductase (CAR) from Mycobacterium marinum and native alcohol dehydrogenase Adh5 (figure 2), this led to production of up to $120 \, \text{mg} \, \text{l}^{-1}$ fatty alcohols in shake flasks [19]. In another study, after expression of a high-activity heterologous fatty acid reductase (FAR) (figure 2), blocking competing pathways by deletion of DGH1, HFD1 and ADH6 together with limiting NADPH and carbon usage by deleting glutamate dehydrogenase encoded by GDH1, a strain producing $1.2 \, \text{g} \, \text{l}^{-1}$ fatty alcohols in shake flasks was obtained [58]. However, impaired growth occurred due to intracellular fatty alcohol accumulation. The expression of the multi-functional transporter FATP1 from human in a fatty alcohol producing yeast strain was shown to facilitate

royalsocietypublishing.org/journal/rsob    Open Biol. 9: 190049

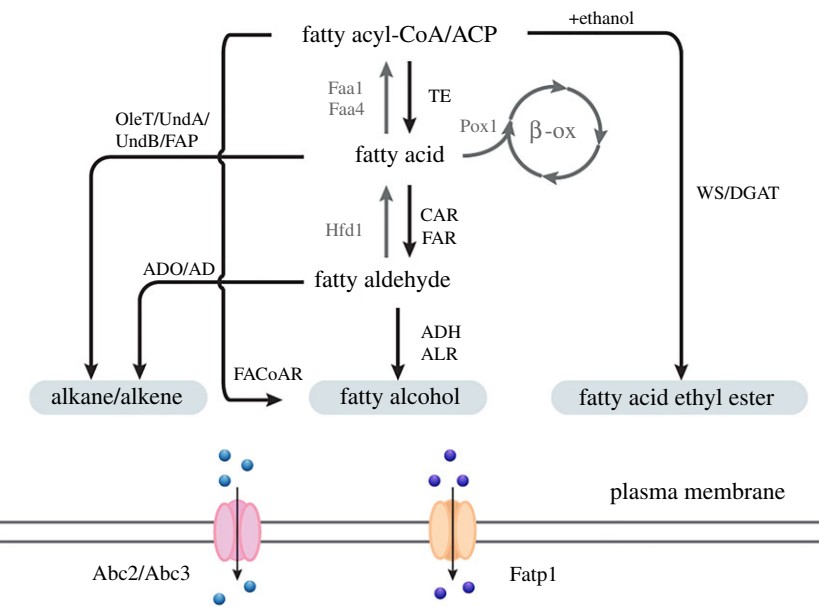

**Figure 2.** Heterologous pathways for fatty acid-derived biofuel synthesis and secretion. The arrows in grey represent the steps involved in fatty acid degradation in yeast. Faa1/4, fatty acyl-CoA synthetases; Pox1, fatty acyl-CoA oxidase; TE, thioesterase; Hdf1, aldehyde dehydrogenase; CAR, carboxylic acid reductase; FAR, fatty acid reductase; ADH, alcohol dehydrogenase; ALR, aldehyde reductase; ADO, aldehyde deformylating oxygenase; AD, aldehyde decarbonylase; WS/DGAT, wax ester synthase/acyl-CoA: diacylglycerol acyltransferase; FAP, fatty acid photodecarboxylase; Abc2/3, alkane transporter from *Yarrowia lipolytica*; Fatp1, mammalian fatty alcohol transporter; β-ox, β-oxidation.

fatty alcohol export (figure 2), which benefited production levels as well as the cell fitness, and resulted in 4.5-fold more extracellular fatty alcohols than the control strain [59]. Based on a yeast strain producing S/MCFAs, 1-octanol was successfully generated with the two-step pathway via *M. marinum* CAR and aldehyde reductase Ahr from *Escherichia coli*, demonstrating that the chain length specificity of FAS is the decisive factor for producing fatty alcohols of a specific chain length [60].

## 4.2. Alka(e)nes

Alkenes and alkanes serve as the major constituents of gasoline, diesel and jet fuel. Several alka(e)ne biosynthesis pathways have been successfully demonstrated in microbes in recent years [18,61–63]. Fatty acids, fatty acyl-CoA/ACP and fatty aldehydes are the major precursors that can be used to generate alka(e)nes via corresponding enzymes. Both aldehyde deformylating oxygenase (ADO) and aldehyde decarbonylase (AD) use fatty aldehydes as substrate to facilitate the formation of alkanes (figure 2). ADO is a non-haem di-iron oxygenase requiring molecular oxygen and an external reducing system to provide four electrons, yielding hydrogen peroxide ($H_2O_2$) and formate as by-products [64] (figure 2). The well-known ADs, *Drosophila melanogaster* CYP4G1 and *Arabidopsis* CER1, are naturally involved in long-chain alkane biosynthesis and have been successfully expressed in *S. cerevisiae* to generate $C_{n-1}$ alkanes from $C_n$ fatty aldehydes [62]. Fatty acid decarboxylases can catalyse the one-step decarboxylation from $C_n$ fatty acids to $C_{n-1}$ 1-alkenes in a process that avoids the formation of fatty aldehydes as intermediates. OleT was reported as a cytochrome P450 enzyme that is responsible for the conversion of C12–C20 fatty acids to corresponding 1-alkenes using $H_2O_2$ as the sole electron and oxygen donor [65] (figure 2). In addition, UndA and UndB were identified as fatty acid decarboxylases for medium-chain 1-alkene synthesis that specifically convert C10–C14

fatty acids and C10–C16 fatty acids, respectively (figure 2). Fatty acyl-CoA/ACPs as substrates can be used to synthesize long-chain alkenes by olefin synthase, a multi-domain polyketide synthase (PKS) from cyanobacteria, through an elongation/sulfonation/decarboxylation mechanism [66]. Furthermore, an algal fatty acid photodecarboxylase (FAP) driven by light was recently found which can convert fatty acids to corresponding alka(e)nes, and it was successfully expressed in *E. coli* to generate hydrocarbons in the presence of visible light [67] (figure 2).

The implementation of alka(e)ne biosynthesis in yeast has made significant progress during recent years. However, the low efficiency of pathway enzymes and the strong competition of fatty alcohol accumulation for metabolic precursors and intermediates are considered the major obstacles of further alka(e)ne production improvement in *S. cerevisiae* [14,68]. Therefore, compartmentalization in yeast organelles turned out to be a promising strategy that provides a suitable environment for alka(e)ne production via isolating the synthesis pathway from the competing pathways in the cytosol. Peroxisomes represent a suitable location for alka(e)ne synthesis not only because of the absence of ALRs/ADHs, but also because of the potential NADPH supply from the peroxisomal NADP-dependent isocitrate dehydrogenase isoenzyme Idp3 [69]. Recently, the alkane synthesis pathway consisting of *Synechococcus elongatus* ADO (*Se*ADO) together with *Mycobacterium marinum* CAR (*Mm*CAR) was targeted to the peroxisomes in yeast, yielding around 0.12 mg l$^{-1}$ alkanes, which was a 90% higher alkane titre than that yielded by the cytosolic pathway [14]. After further increasing the precursor supply in the peroxisomes and deleting the cytosolic ALR/ADH genes *ADH5* and *SFA1*, alkane production reached 1.2 mg l$^{-1}$ with significantly decreased fatty alcohol accumulation. Moreover, an additional study focusing on medium-chain alkane synthesis in yeast indicates that the compartmentalization in peroxisomes could work as an efficient strategy in this context as well [70].

royalsocietypublishing.org/journal/rsob  Open Biol. 9: 190049

The toxicity caused by the accumulation of alka(e)nes negatively affects the cell growth and limits the production yield. Consequently, some studies towards improving solvent tolerance were conducted in yeast. The native plasma membrane efflux pumps Snq2 and Pdr5 were identified in *S. cerevisiae* as contributing to alkane export and tolerance by reducing intracellular levels, specifically for C10 and C11 alkanes [71]. The heterologous transporters Abc2 and Abc3 from *Yarrowia lipolytica* significantly increased tolerance against decane and undecane in *S. cerevisiae* through maintaining lower intracellular alkane levels [72] (figure 2). Furthermore, mammalian FATP1 previously identified as a fatty alcohol exporter was expressed in yeast to benefit 1-alkene secretion. The implementation of dynamic regulation, through expressing PfUndB under the control of the *GAL7* promoter with deletion of *GAL80* to separate the cell growth and production process and replacing the electron transfer system by the NADH-based putidaredoxin reductase system, finally enabled a yeast cell factory to produce 35.3 mg l$^{-1}$ 1-alkenes with more than 80% being secreted, which is a 10-fold improvement compared with earlier reported hydrocarbon production by *S. cerevisiae* [63,73].

## 4.3. Fatty acid ethyl esters

The biosynthesis of FAEEs, considered as potential diesel fuel replacement, was demonstrated in yeast. Ethanol and acyl-CoAs are the essential precursors involved in FAEE synthesis that can be catalysed by a wax ester synthase/acyl-CoA: diacylglycerol acyltransferase (WS/DGAT) (figure 2). Generally, most WSs naturally accept acyl groups with a chain length of C16 or C18 and linear alcohols with a chain length ranging from C12 to C20, and various WSs have different substrate chain length preferences [17]. Five heterologous WSs from bacteria and mammals were expressed and evaluated in *S. cerevisiae* to investigate their substrate preferences [17]. The results showed that the WS from *Marinobacter hydrocarbonoclasticus* had the best performance using ethanol as the substrate *in vitro* compared with the other enzymes, and enabled a titre of 6.3 mg l$^{-1}$ FAEEs after expressing it in the engineered yeast strain.

In order to establish a stable expression system, the heterologous wax ester synthase gene (*ws2*) was integrated into the yeast chromosomes in multiple copies, resulting in an increase in FAEE production of up to 34 mg l$^{-1}$ [74]. Subsequently, the endogenous acyl-CoA binding protein and a bacterial NADP$^{+}$-dependent GAPN were overexpressed in the integration strain to enhance the precursor and cofactor supply, which enabled a further 40% increase in FAEE production. During the synthesis of FAEEs in yeast, the concentrations of ethanol and acyl-CoA influence the yield of the final product. Thus, the carbon flux was redirected towards acetyl-CoA, the precursor of acyl-CoA, by overexpressing the alcohol dehydrogenase (*ADH2*), acetaldehyde dehydrogenase (*ALD6*) and ACS encoded by heterologous gene *acs$_{SE}^{L641P}$*, together with the integrated *ws2*, resulting in a threefold improvement [39]. Then, the PHK pathway was introduced to enhance acetyl-CoA supply by heterologous expression of *xpkA* and either *ack* from *Aspergillus nidulans* or *pta* from *Bacillus subtilis*. Both PHK pathways helped to generate around 5.0 mg g$^{-1}$ cell dry weight FAEEs, an up to a 1.7-fold improvement. Besides, reducing the competition of other pathways for acyl-CoA also permits an overproduction of FAEEs. Therefore,

by eliminating the formation of steryl esters (SEs) and triacylglycerols (TAGs), a threefold increase in FAEE production was achieved [75]. The heterologous expression of a type I FAS from *Brevibacterium ammoniagenes* coupled with WS/DGAT yielded a 6.3-fold increase in FAEE production compared with a strain not containing the heterologous FAS [45]. Additionally, the alternative carbon source, glycerol, with the advantage of being a low-price and highly reduced substrate was used to produce FAEEs in *S. cerevisiae* [76]. The titre of FAEEs reached 0.52 g l$^{-1}$ after increasing the ethanol formation from glycerol, blocking the glycerol export route and adding exogenous fatty acids, which is the highest reported FAEE production to date in yeast.

## 5. Perspectives

Progress in developing more advanced biotechnology tools has led to more efficient engineering of microbes. An example is the clustered regularly interspaced short palindromic repeats (CRISPR)/Cas technology, which allows fast multiplex genome editing and has significantly shortened the time required for strain construction. Even though there has been much progress on engineering yeast for production of advanced biofuels in the laboratory, it is still challenging to meet the titre, rate and yield (TRY) requirements for commercial production of low-value fatty acid-derived products (table 1). In order to meet the commercial requirements, the yields and productivities of laboratory-scale processes need to approach around 85% of the theoretically possible yield and the fermentation has to be scaled up drastically [82]. The scientific progress, however, lays the basis for further development in case some of the key barriers can be passed [5].

Owing to the relatively low value of many fatty acid-derived chemicals, improving the utilization of the carbon source is a promising strategy that contributes to high TRY metrics. Therefore, attempts towards the utilization of single carbon feedstock such as CO$_2$ and methane have attracted much attention, and were shown to be a feasible alternative with high carbon- and energy-conversion efficiency [83]. However, there are many challenges that need to be conquered before the application of this concept in industrial production can be realized. For example, the carbon atom from CO$_2$ possesses a high oxidation state that requires large amounts of reducing power in microbes to efficiently remove the oxygen atoms for it to be used for hydrocarbon synthesis. When considering methane-based metabolism, the activation of the C–H bond in methane is a costly and extremely inefficient process, which indicates that a more feasible and efficient design is needed for methane utilization by microorganisms [84].

In many cases, the poor performance of the key enzymes involved in the different biosynthetic pathways is the major obstacle for improving product formation. This is, for example, the case for ADO, which has low catalytic activity even in its native host [85]. In addition, expression of heterologous membrane proteins is usually challenging owing to the distinct membrane structures between organisms [86,87], and poor expression is often observed for the members of the superfamily of cytochrome P450 enzymes such as OleT with one of potential reasons being cofactor (haem) deficiency [65]. Protein engineering of these enzymes can serve as a feasible strategy to enable further enhancement of final product levels through a

**Table 1.** Comparison of biofuel production from different organisms. FAAs, free fatty acids; OCFAs, odd chain fatty acids; FOHs, fatty alcohols; SCAs, short chain alkanes; FAEEs, fatty acid ethyl esters; VLCFOHs, very-long-chain fatty alcohols; YNB, yeast nitrogen base; SD, synthetic defined.

| microorganisms | product | titre (g l$^{-1}$) | yield$^a$ (g g$^{-1}$) | medium | cultivation condition | reference |
|---|---|---|---|---|---|---|
| S. cerevisiae | FAAs | 1.0 | 0.05 | MM$^b$ | shake flask | [19] |
| E. coli | FAAs | 3.9 | N.C.$^c$ | MK$^d$ | fed-batch | [77] |
| E. coli | FAAs | 1.2 | 0.06 | MM | shake flask | [53] |
| Y. lipolytica | OCFAs | 0.75 | N.C. | YNB | fed-batch | [78] |
| S. cerevisiae | FOHs | 0.12 | 0.004 | MM | shake flask | [19] |
| S. cerevisiae | VLCFOHs | 0.084 | 0.0028 | MM | shake flask | [52] |
| S. cerevisiae | FOHs | 0.1 | 0.005 | MM | shake flask | [16] |
| E. coli | FOHs | 1.7 | 0.028 | MM | batch | [79] |
| S. cerevisiae | 1-alkenes | 0.035 | 0.0011 | MM | shake flask | [63] |
| Y. lipolytica | alkanes | 0.023 | N.C. | YNB | shake flask | [80] |
| E. coli | SCAs | 0.58 | N.C. | MR$^d$ | fed-batch | [81] |
| S. cerevisiae | FAEEs | 0.034 | N.C. | SD medium | shake flask | [74] |
| Y. lipolytica | FAEEs | 0.14 | N.C. | YNB | shake flask | [80] |

$^a$Yield was defined as carbon source conversion rate to biofuel production.
$^b$MM, minimal medium.
$^c$Unable to calculate because of the complex composition of the medium.
$^d$Minimal medium with extra yeast extract.

more efficient metabolite conversion. Thus, with protein engineering strategies such as rational design based on known or simulated protein structures or random approaches through directed evolution, the cofactor, substrate or product specificity can be altered and benefit the overall improvement of enzyme activities. A successful example is the engineering of a type I FAS to generate more S/MCFAs in yeast, which offers the possibility to alter the fatty acid chain length and enables a larger diversity of biofuels produced by engineered strains [47]. However, gaining additional knowledge on enzymes and their structures as well as the development of advanced tools used for *in silico* analysis of proteins require more effort and attention in future studies.

With still relatively low yields and rates in biofuel production, the tolerance against toxic chemicals and fermentation processes needs to be further improved in order to achieve high TRY metrics to meet the requirements of commercial application [82,88]. Several successful attempts have shown that identifying respective transporters can be a feasible approach to releasing growth inhibition in microbial cell factories [59,63,72]. In addition, ALE towards toxic chemicals and inhibitory conditions has proven to be a promising method that can benefit the production of biofuels and other molecules, for instance resistance against high temperature or oxidative stress could be obtained by ALE [89,90]. If product formation can be linked to cell growth or survival, it is also possible to evolve strains for increased production. In this way, carotenoid production was improved by hydrogen peroxide-challenged adaptive evolution [91].

In addition, a number of biosensors have been developed to detect specific molecules and have been recently employed to facilitate the production of some valuable compounds, such as using a malonyl-CoA sensor to improve the production of 3-hydroxypropionic acid and fatty acids [77,92–94]. Nevertheless, the number of metabolites detectable as well as the properties (e.g. specificity) of existing biosensors need to be improved for additional applications [95]. Recently, the concept of synthetic product addiction facilitated by biosensors was proposed as a promising solution that would benefit high-yield bio-manufacturing [96]. Through linking production of the desired metabolite to the expression of non-conditionally essential genes, the product-addicted strains with biosynthetic capacity in the population will be selected without constraining the medium, thus providing production stability over many generations. With the help of such advanced biotechnology tools, efforts in the coming years will focus on how to improve the TRY metrics in order to meet the commercial requirement for lower value products and the production of higher value molecules using microorganisms.

Data accessibility. This article does not contain any additional data.

Authors' contributions. Y.H. outlined and drafted the manuscript; Z.Z., V.S. and J.N. revised the manuscript. All authors gave their final approval before submission.

Competing interests. We declare we have no competing interests.

Funding. This work was funded by the Swedish Foundation for Strategic Research, the Novo Nordisk Foundation (NNF10CC1016517) and the Knut and Alice Wallenberg Foundation.

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
