## [Reviewer comments · Open Biology]

Review History

RSOB-19-0049.R0 (Original submission)

Review form: Reviewer 1

Recommendation

Accept with minor revision (please list in comments)

Are each of the following suitable for general readers?

- a) **Title**
Yes
- b) **Summary**
Yes
- c) **Introduction**
Yes

Is the length of the paper justified?

Yes

Should the paper be seen by a specialist statistical reviewer?

Yes

Is it clear how to make all supporting data available?

Not Applicable

Is the supplementary material necessary; and if so is it adequate and clear?

Not Applicable

Do you have any ethical concerns with this paper?

No

Comments to the Author

The authors compose a review article on the topic of engineering yeast (specifically, *S. cerevisiae*) for the production of molecules related to fatty acid production. While the publication and topic is timely for the field, there is a bit that can be improved for the delivery to better serve the field. These are enumerated in the following points of consideration for the authors:

1. At the onset, the overall manuscript is not well balanced with nearly 1/3 of the references coming from the authors of this paper. Immediately, this presents a bias in the article and does not reflect well on the field as a whole given the work in this area. There are certainly many either additional or alternative papers that should be cited as well.
2. The prospectives section was a bit weak and vague and not a great ending for the paper. Simply stating the obvious (such as the fact that CRISPR exists that better enzymes need to be found for improving TRY) does not lead the reader to gain much perspective. A more meaningful section should be included instead.
3. The premise that *S. cerevisiae* is used in ethanol production and thus will be a good host for fatty acid production as it is well studied is a bit ill placed here. The authors should better contextualize TRY values found here with other host organisms. This is true, especially when values of the low mg/L are presented for molecules that are reported to be used as fuels – that will require much higher titers.
4. The figures are rather simplified biochemical pathways that neglect both the regulation inherent in these pathways as well as the fact that they do not showcase any of the “engineering” aspect that is claimed in the title.

Review form: Reviewer 2

Recommendation

Accept with minor revision (please list in comments)

Are each of the following suitable for general readers?

- a) **Title**
Yes

b) Summary

Yes

c) Introduction

Yes

Is the length of the paper justified?

Yes

Should the paper be seen by a specialist statistical reviewer?

No

Is it clear how to make all supporting data available?

Not Applicable

Is the supplementary material necessary; and if so is it adequate and clear?

Not Applicable

Do you have any ethical concerns with this paper?

No

Comments to the Author

"Engineering *Saccharomyces cerevisiae* cells for production of fatty acid derived biofuels and chemicals" by Hu et al

Synopsis:

It is a very informative review about the advances in metabolic engineering of yeast for biofuels production. The focus is restricted on using glucose/starch based feedstock and *S. cerevisiae*, other carbon sources and organisms are not covered.

The introduction is separated into two parts, the first introducing the carbon flux and possibilities in the yeast cell from sugar to the desired product, and the second part deals with the redox balance in yeast, which is critical for the production of the highly reduced target compounds.

Native FA biosynthesis in yeast cells is introduced and possibilities of manipulating the FAS complexes are presented, together with strategies and examples to enhance and change the FA product spectrum by incorporating heterologous genes into yeast. The same is then applied to other products, ordered by their chemical profile: fatty alcohols, alkanes/alkenes and fatty acid esters. The final part "perspectives" presents further strategies and examples and highlights recent developments.

The introduction and the main parts of the manuscript are overall very well written and understandable and appear to cover almost everything essential from recent developments. Only the "perspectives" part appears a bit shallow and provides neither a refined summary nor a clear perspective for researchers in the field. A lot of new and exciting developments are only very briefly mentioned with one or two references. Here, the authors should expand the focus a bit from pure biofuel compounds to other fermentation products from yeast to find examples for successful application of new concepts. Recent strategies such as biosensors or "product addiction" are not included.

minor revisions:

a) page 5 line 6: (ACL) was overexpressed

b) page 5 line 11: malate synthase

c) page 12, lines 1 to 4: "implementation of dynamic regulation" this should be elaborated in a few sentences. ALE is mentioned briefly, and could also be elaborated more. A connection to other fermentation products should be made if no biofuels example is available. As a suggestion, "biosensors" in combination with HTS strategies or "product addiction" could be mentioned as a perspective, if no examples for application in biofuel production are available. They have been successfully applied to other low- or medium value products.

d) page 14, lines 10 to 11: "In many cases, the poor performance of the key enzymes involved in the different biosynthetic pathways is the major obstacle for improving product formation." - this is a very true statement for all heterologous production attempts in yeast. Some examples and possible explanations/reasons should be provided

e) page 14, line 23: "to meet the requirement of commercial application." A brief comparison of the current biofuel-performance of yeast with other relevant host platforms should be given. Absolute numbers and yields show up here and there in the manuscript, but summarized and compared they would provide more value to the reader.

f) page 14, line 25: "In addition, adaptive laboratory evolution (ALE) towards toxic chemicals is a method that can benefit production of biofuels and other molecules" - see comment c)

Decision letter (RSOB-19-0049.R0)

10-Apr-2019

Dear Dr Nielsen

We are pleased to inform you that your manuscript RSOB-19-0049 entitled "Engineering *Saccharomyces cerevisiae* cells for production of fatty acid derived biofuels and chemicals" has been accepted by the Editor for publication in Open Biology. The reviewer(s) have recommended publication, but also suggest some minor revisions to your manuscript. Therefore, we invite you to respond to the reviewer(s)' comments and revise your manuscript.

Please submit the revised version of your manuscript within 7 days. If you do not think you will be able to meet this date please let us know immediately and we can extend this deadline for you.

When submitting your revised manuscript, you will be able to respond to the comments made by the referee(s) and upload a file "Response to Referees" in "Section 6 - File Upload". You can use this to document any changes you make to the original manuscript. In order to expedite the

processing of the revised manuscript, please be as specific as possible in your response to the referee(s).

- 1) A text file of the manuscript (doc, txt, rtf or tex), including the references, tables (including captions) and figure captions. Please remove any tracked changes from the text before submission. PDF files are not an accepted format for the "Main Document".
- 2) A separate electronic file of each figure (tiff, EPS or print-quality PDF preferred). The format should be produced directly from original creation package, or original software format. Please note that PowerPoint files are not accepted.
- 3) Electronic supplementary material: this should be contained in a separate file from the main text and meet our ESM criteria (see <http://royalsocietypublishing.org/instructions-authors#question5>). All supplementary materials accompanying an accepted article will be treated as in their final form. They will be published alongside the paper on the journal website and posted on the online figshare repository. Files on figshare will be made available approximately one week before the accompanying article so that the supplementary material can be attributed a unique DOI.

Online supplementary material will also carry the title and description provided during submission, so please ensure these are accurate and informative. Note that the Royal Society will not edit or typeset supplementary material and it will be hosted as provided. Please ensure that the supplementary material includes the paper details (authors, title, journal name, article DOI). Your article DOI will be 10.1098/rsob.2016[*last 4 digits of e.g. 10.1098/rsob.20160049*].

- 4) A media summary: a short non-technical summary (up to 100 words) of the key findings/importance of your manuscript. Please try to write in simple English, avoid jargon, explain the importance of the topic, outline the main implications and describe why this topic is newsworthy.

Images

Data-Sharing

It is a condition of publication that data supporting your paper are made available. Data should be made available either in the electronic supplementary material or through an appropriate repository. Details of how to access data should be included in your paper. Please see <http://royalsocietypublishing.org/site/authors/policy.xhtml#question6> for more details.

Data accessibility section

Sincerely,

The Open Biology Team
mailto:openbiology@royalsociety.org

Reviewer(s)' Comments to Author:

Referee: 1

Comments to the Author(s)

The authors compose a review article on the topic of engineering yeast (specifically, *S. cerevisiae*) for the production of molecules related to fatty acid production. While the publication and topic is timely for the field, there is a bit that can be improved for the delivery to better serve the field. These are enumerated in the following points of consideration for the authors:

1. At the onset, the overall manuscript is not well balanced with nearly 1/3 of the references coming from the authors of this paper. Immediately, this presents a bias in the article and does not reflect well on the field as a whole given the work in this area. There are certainly many either additional or alternative papers that should be cited as well.
2. The perspectives section was a bit weak and vague and not a great ending for the paper. Simply stating the obvious (such as the fact that CRISPR exists that better enzymes need to be found for improving TRY) does not lead the reader to gain much perspective. A more meaningful section should be included instead.
3. The premise that *S. cerevisiae* is used in ethanol production and thus will be a good host for fatty acid production as it is well studied is a bit ill placed here. The authors should better contextualize TRY values found here with other host organisms. This is true, especially when values of the low mg/L are presented for molecules that are reported to be used as fuels – that will require much higher titers.
4. The figures are rather simplified biochemical pathways that neglect both the regulation inherent in these pathways as well as the fact that they do not showcase any of the “engineering” aspect that is claimed in the title.

Referee: 2

Comments to the Author(s)

"Engineering *Saccharomyces cerevisiae* cells for production of fatty acid derived biofuels and chemicals" by Hu et al

Synopsis:

It is a very informative review about the advances in metabolic engineering of yeast for biofuels production. The focus is restricted on using glucose/starch based feedstock and *S. cerevisiae*, other carbon sources and organisms are not covered.

The introduction is separated into two parts, the first introducing the carbon flux and possibilities in the yeast cell from sugar to the desired product, and the second part deals with the redox balance in yeast, which is critical for the production of the highly reduced target compounds.

Native FA biosynthesis in yeast cells is introduced and possibilities of manipulating the FAS complexes are presented, together with strategies and examples to enhance and change the FA product spectrum by incorporating heterologous genes into yeast. The same is then applied to other products, ordered by their chemical profile: fatty alcohols, alkanes/alkenes and fatty acid esters. The final part "perspectives" presents further strategies and examples and highlights recent developments.

The introduction and the main parts of the manuscript are overall very well written and understandable and appear to cover almost everything essential from recent developments. Only the "perspectives" part appears a bit shallow and provides neither a refined summary nor a clear perspective for researchers in the field. A lot of new and exciting developments are only very briefly mentioned with one or two references. Here, the authors should expand the focus a bit from pure biofuel compounds to other fermentation products from yeast to find examples for successful application of new concepts. Recent strategies such as biosensors or "product addiction" are not included.

minor revisions:

a) page 5 line 6: (ACL) was overexpressed

b) page 5 line 11: malate synthase

c) page 12, lines 1 to 4: "implementation of dynamic regulation" this should be elaborated in a few sentences. ALE is mentioned briefly, and could also be elaborated more. A connection to other fermentation products should be made if no biofuels example is available. As a suggestion, "biosensors" in combination with HTS strategies or "product addiction" could be mentioned as a perspective, if no examples for application in biofuel production are available. They have been successfully applied to other low- or medium value products.

d) page 14, lines 10 to 11: "In many cases, the poor performance of the key enzymes involved in the different biosynthetic pathways is the major obstacle for improving product formation." - this is a very true statement for all heterologous production attempts in yeast. Some examples and possible explanations/reasons should be provided

e) page 14, line 23: "to meet the requirement of commercial application." A brief comparison of the current biofuel-performance of yeast with other relevant host platforms should be given. Absolute numbers and yields show up here and there in the manuscript, but summarized and compared they would provide more value to the reader.

f) page 14, line 25: "In addition, adaptive laboratory evolution (ALE) towards toxic chemicals is a method that can benefit production of biofuels and other molecules" - see comment c)

Decision letter (RSOB-19-0049.R1)

25-Apr-2019

Dear Dr Nielsen

We are pleased to inform you that your manuscript entitled "Engineering *Saccharomyces cerevisiae* cells for production of fatty acid derived biofuels and chemicals" has been accepted by the Editor for publication in Open Biology.

Sincerely,

The Open Biology Team
mailto: openbiology@royalsociety.org